

# Establishing Silphids in the invertebrate DNA toolbox: a proof of concept

Katherine E. Higdon, Kristin E. Brzeski, Melanie A. Ottino and Tara L. Bal

College of Forest Resources and Environmental Science, Michigan Technological University, Houghton, MI, United States of America

## ABSTRACT

Environmental DNA (eDNA) analyses are an increasingly popular tool for assessing biodiversity. eDNA sampling that uses invertebrates, or invertebrate DNA (iDNA), has become a more common method in mammal biodiversity studies where biodiversity is assessed via diet analysis of different coprophagous or hematophagous invertebrates. The carrion feeding family of beetles (*Silphidae: Coleoptera*, Latreille (1807)), have not yet been established as a viable iDNA source in primary scientific literature, yet could be useful indicators for tracking biodiversity in forested ecosystems. Silphids find carcasses of varying size for both food and reproduction, with some species having host preference for small mammals; therefore, iDNA Silphid studies could potentially target small mammal communities. To establish the first valid use of iDNA methods to detect Silphid diets, we conducted a study with the objective of testing the validity of iDNA methods applied to Silphids using both Sanger sequencing and high throughput Illumina sequencing. Beetles were collected using inexpensive pitfall traps in Alberta, Michigan in 2019 and 2022. We successfully sequenced diet DNA and environmental DNA from externally swabbed Silphid samples and diet DNA from gut dissections, confirming their potential as an iDNA tool in mammalian studies. Our results demonstrate the usefulness of Silphids for iDNA research where we detected species from the genera *Anaxyrus, Blarina, Procyon, Condylura, Peromyscus, Canis,* and *Bos*. Our results highlight the potential for Silphid iDNA to be used in future wildlife surveys.

## INTRODUCTION

The growing use of genetic tools has fostered discussions around the application and use of environmental DNA (eDNA) or invertebrate DNA (iDNA) methods in biodiversity surveys (*Abrams et al., 2019*). New iDNA tools are used in conservation biology as complementary tools to traditional vertebrate trapping techniques where conservation practitioners can now also trap invertebrates to gain data on vertebrates through genomic methods (*Bohmann, Schnell & Gilbert, 2013*). For instance, blood-feeding and dung-feeding insects can be used to estimate vertebrate wildlife diversity through metabarcoding and sequencing their diet. Currently, methods have been developed that use mosquitoes and flies (Diptera), ticks (Ixodia), leeches (Hirudinea), dung beetles (Scarabidae) and other invertebrate species to indirectly study wildlife (*Kent & Norris, 2005*; *Gariepy et al., 2012*; *Lee, Sing & Wilson,*

Corresponding author
Tara L. Bal, tlbal@mtu.edu

*2015*; *Schnell et al., 2015*; *Drinkwater et al., 2021*), but to our knowledge, no studies have yet demonstrated extracting, amplifying, or sequencing iDNA from *Silphidae* (Silphids: Coleoptera) Latreille (1807). Silphids, composing two sub-family groups commonly named burying or sexton beetles (*Nicrophorinae*) and carrion beetles (*Silphinae*), use and feed on the carcasses of animals, with burying beetles specializing on small mammal or small avian carcasses and burying them as their name suggests. While Silphids have been suggested in literature as a potential iDNA source, previous work has not demonstrated the actual use of these species for biodiversity studies or wildlife monitoring (*Calvignac-Spencer et al., 2013*). Our research objective was to use small scale sampling to determine the applicability of Silphids for iDNA surveys and their potential use as a tool for monitoring small mammal diversity and thus create a proof-of-concept methodological approach for large-scale studies.

Silphids, specifically *Nicrophorinae*, may be particularly useful for small mammal biodiversity surveys as they favor small carcasses for both feeding and reproductive purposes (*Scott, 1998*), while other viable iDNA invertebrates might not exclusively favor small-bodied carcasses (*Kent & Norris, 2005*; *Gariepy et al., 2012*; *Lee, Sing & Wilson, 2015*; *Schnell et al., 2015*; *Drinkwater et al., 2021*). This has the potential to be especially beneficial as capture-based small mammal studies often struggle with trap biases associated with trap sizes which target only certain size or types of small mammals, limiting the ability to detect the full diversity of small mammals present (*Anthony et al., 2005*). Silphids can be found across the northern hemisphere and are fairly ubiquitous, consisting of about 75 species in the genus *Nicrophorus* alone (*Anderson, 1982*; *Scott, 1998*). *Silphinae*, or the carrion beetles, tend to be less particular in requirements for the size of carrion, being found on all sizes of carcasses, and consist of more than 100 extant species (*Anderson & Peck, 1985*; *Peck & Miller, 1993*). Different species vary in seasonal patterns, habitat specificity, and diet preferences, but overall, excluding the one federally listed endangered species (the American burying beetle; *Nicrophorus americanus* Olivier), Silphids are generally regarded as widely distributed, common insects that feed on a variety of carrion (*Anderson & Peck, 1985*; *Scott, 1998*).

The variation in Silphid natural ecologies may be helpful in exploring their potential as an iDNA tool beyond monitoring, including potentially deeper insights into faunal diversity and density, but especially with regards to mortality information of their hosts (*Calvignac-Spencer et al., 2013*). To reduce competition, different Silphid species have niche partitioned by selecting different size carcasses and through variation in seasonal activity patterns and habitat preferences (*Anderson, 1982*). For example, *Von Hoermann et al. (2018)* found differences in abundance of specific Silphids between forest stands with varying management intensities, however overall Silphid abundance was not affected. Though they report fewer and less diverse Silphid species in urban forests *versus* rural forests, *Wolf & Gibbs (2004)* suggest burying beetles in urban forests did not appear any more impoverished than those in rural forests. Findings like these suggest that though there may be specialist Silphids that prefer specific habitats or carcass types, they do appear to be abundant enough in terrestrial systems across the northern hemisphere for biomonitoring applications as a source of iDNA. We sought to establish what methods were needed to

successfully detect iDNA in Silphid diets by conducting a proof-of-concept study using Sanger sequencing methods to first validate the process, then expand sampling to use more comprehensive high throughput sequencing metabarcoding methods while optimizing the volumes of reagents needed for successful polymerase chain reaction (PCR) amplification for contemporary iDNA biodiversity surveys.

## MATERIALS & METHODS

### Study area

We conducted this study in Michigan's Upper Peninsula in Baraga County near the village of Alberta, Michigan on Michigan Technological University's Ford Forest (46°37′40.7″N, 88°28′31.5″W). Study sites were typical of Great Lakes Northern Hardwoods forests. Average annual precipitation for Alberta, MI is 88.9 cm and average snowfall annually is approximately 390.1 cm, with average temperatures of 4.7 °C with a range from −10.8 °C to 18.1 °C (*Hupperts et al., 2020*). These sites were harvested with varying intensity, ranging from small clearcuts to single tree selection, two and five years prior to sampling, in 2017. The goal of this sampling was to demonstrate the utility of beetles for iDNA methods, thus assessing trends associated with trapping locations or species preferences was outside the scope of this study. Yet all traps were set in areas with forest cover or complete ground cover with mixed herbaceous species and woody vegetation.

### Field methods

We collected samples in the summers of 2019 and 2022. The 2019 samples were used to develop and refine field and lab methods, where we improved both field sampling and lab sequencing, as outlined below. To trap beetles, we used homemade pitfall traps, which were 473 mL plastic cups buried in the ground with the opening level with the soil surface and a lid (a foam plate) covering it with a 16 mm gap from the ground, with bait in the bottom. We experimented with trap bait to find an attractive lure out of those suggested in literature that would limit the potential for DNA contamination within traps. The four bait types used were Danny King's Catfish Punchbait® (*Andrew, 2016*), raw ground beef, frozen pinky mice (pet food), and raw chicken. In June and July of 2019, traps were set in pairs approximately ten meters apart, one being baited with catfish bait and the other with raw ground beef. In the last three weeks of trapping during September and October, we switched to pinky mice (universal pet store food) being used exclusively in the paired traps with the objective of attracting more burying beetles (*Nicrophorus* sp.) as we had yet to capture one (*Andrew, 2016*). Bait was automatically replenished or the trap was reset at two weeks but also at one week if there was no bait in the bottom of the trap. For 2022, only raw chicken cut to approximately 16 cm$^3$ was used for bait and replaced if the trap had been disturbed (*Andrew, 2016*).

Our traps also employed a cone and platform configuration of a plastic mesh screen with approximately four by 2 mm holes (Fig. 1), sterilized with bleach prior to being deployed in the field. This was used to separate the relatively large-bodied Silphids from the bait to avoid diet contamination with bait materials. We set traps under as sterile conditions as possible, *e.g.*, gloves were worn during trap assembly and changed in between every
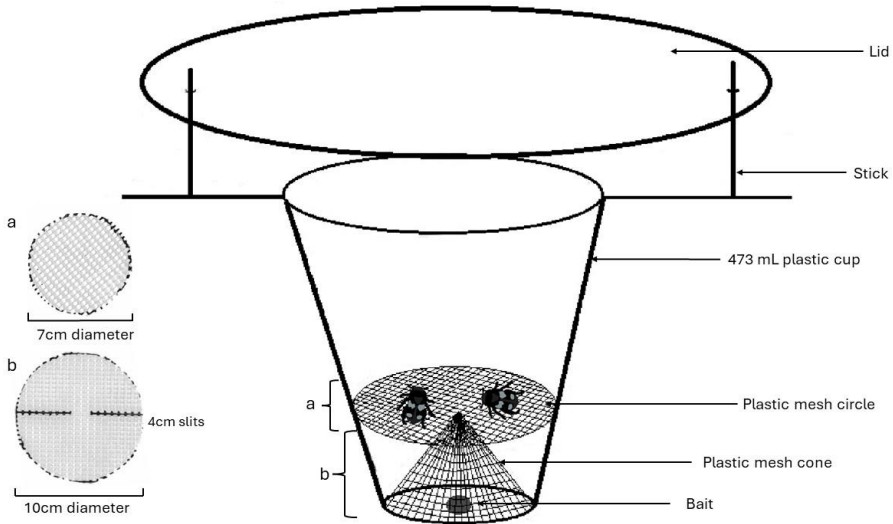

**Figure 1   Model for pitfall trap using a cone and platform configuration of plastic mesh to separate Silphid beetles from bait attractant.** Figure made by Katherine Higdon and Hunter Higdon.

trap or when they came in contact with a potential contaminant. A total of 216 traps set in 108 pairs each ten meters apart and monitored weekly between June 6 and July 27, 2019, beginning once snow had melted and September 14 and October 3, 2019, ending trapping at the first frost. When traps were checked, bait was reset if it was gone (often due to smaller invertebrates going past the mesh layer), and fresh bait was added every two weeks regardless. The mesh component of the trap would occasionally pop out during deployment (likely due to animal disturbance), increasing contamination. Thus, hot glue was used to keep the mesh in its form for the 2022 field season. A total of 151 traps were set starting June 3 through August 21, 2022 for a 12-day period and checked every two to three days so that traps could be reset if disturbed and to increase the chance of getting a viable diet sample.

Silphids were identified to species morphologically since any possible species trapped in Michigan has clear physical differences (*Anderson & Peck, 1985*). Beetles, including larval instars of the carrion beetles, in traps were removed individually using forceps sterilized in the field with alcohol and flame in between each new specimen. With individuals in their own labeled plastic bag, we recorded species and total number of beetles in each trap.

A generalized workflow for both seasons of field, leading into lab methods, is detailed in Fig. 2. For our first season, every individual was sampled for iDNA first by swabbing their exterior, to determine if we could detect vertebrate DNA from the beetle's contact with a vertebrate carcass. Simultaneously during this process, the swabs inevitably collected any excrement, fecal or regurgitate, samples that the beetles produced in the plastic bags. For swabbing in the field, each specimen was swabbed externally for approximately 15 s around the exoskeleton with a sterile, DNA free swab that had been dipped in a tube with a phosphate buffer saline (PBS) solution, prepared in the lab under sterile conditions prior to fieldwork. Swabs were then placed directly into the corresponding labeled tube containing
PBS buffer and put on ice then stored within an hour of collection in a −20 °C freezer until DNA extractions. For the second iDNA collection method, we dissected beetle gut contents for diet DNA in the lab, and all samples were kept frozen until dissections were complete. Gut content of adult Silphid beetles were dissected under sterile lab conditions; larvae were not dissected due to their small size. All dissection tools were autoclaved and gloves were worn at all times when in a fume hood and changed between dissections. Beetles were incised just below the pronotum so that the head and thorax, along with wings, were removed. This left an opening in which a sterile, DNA free swab, was able to be inserted to collect the gut contents. For the second field season, we only collected excrement samples from individuals in the field due to the low success of dissection samples from 2019 and also excluded exoskeleton swabbing from our methods to specifically target diet DNA. Most individuals almost immediately produced an excrement sample once they were in their individual bag, while the remainder produced a sample when slight pressure was applied to their abdomens through lightly pinching their dorsal and ventral sides with care as to not harm them. These samples were collected with swabs and placed in tubes stored on ice in the field and placed in a −20 °C freezer until they could be transported and placed in a −80 °C freezer until DNA extraction. Once a sample was obtained, beetles were gently painted with a bioluminescent, non-toxic paint powder so that a recapture would be noted and not re-swabbed, and then released.

## Molecular methods

All DNA extractions, PCR, and metabarcoding preparation was conducted exclusively in a designated eDNA lab space, with dedicated equipment and rigorous cleaning protocols. We extracted DNA from swabs taken in the field and gut content separately. A general workflow of molecular methods for all sample types is laid out in Fig. 2.

We used Qiagen DNeasy Blood & Tissue protocols with slight volume modifications scaling with the amount of PBS buffer samples were stored in, maintaining the same ratios as recommended in published protocols. For samples collected in 2019, for which we intended to conduct Sanger sequencing to determine if we collected vertebrate DNA, we extracted and processed each beetle independently. For samples collected in 2022, for which we conducted high-throughput sequencing, after we lysed the sample, we pooled up to six samples from the same trap into a spin column for the rest of the DNA extraction. Both DNA extractions included a negative control that was carried through PCR and library preparation.

## Sequencing

*Sanger sequencing*: In 2019 we wanted to determine the best swabbing method and demonstrate we could amplify vertebrate DNA, and therefore, we used Sanger sequencing to identify species. To amplify diet DNA, we used universal vertebrate primers targeting the mitochondrial 12S rRNA vertebrate gene region (*Riaz et al., 2011*). Our PCR reaction was as follows: 10 µL reactions with 1 µL of DNA, 0.4 µL of 5U/ µL AmpliTaq Gold Polymerase, 1 µL of GeneAmp™ 10X Gold Buffer, 0.75 µL of 25 mM MgCl, 0.8 µL of 5 mM dNTPs, 0.2 µL at 10 uM of the forward and reverse vertebrate primer pair 12S-V5-F/R (12S-V5-F:

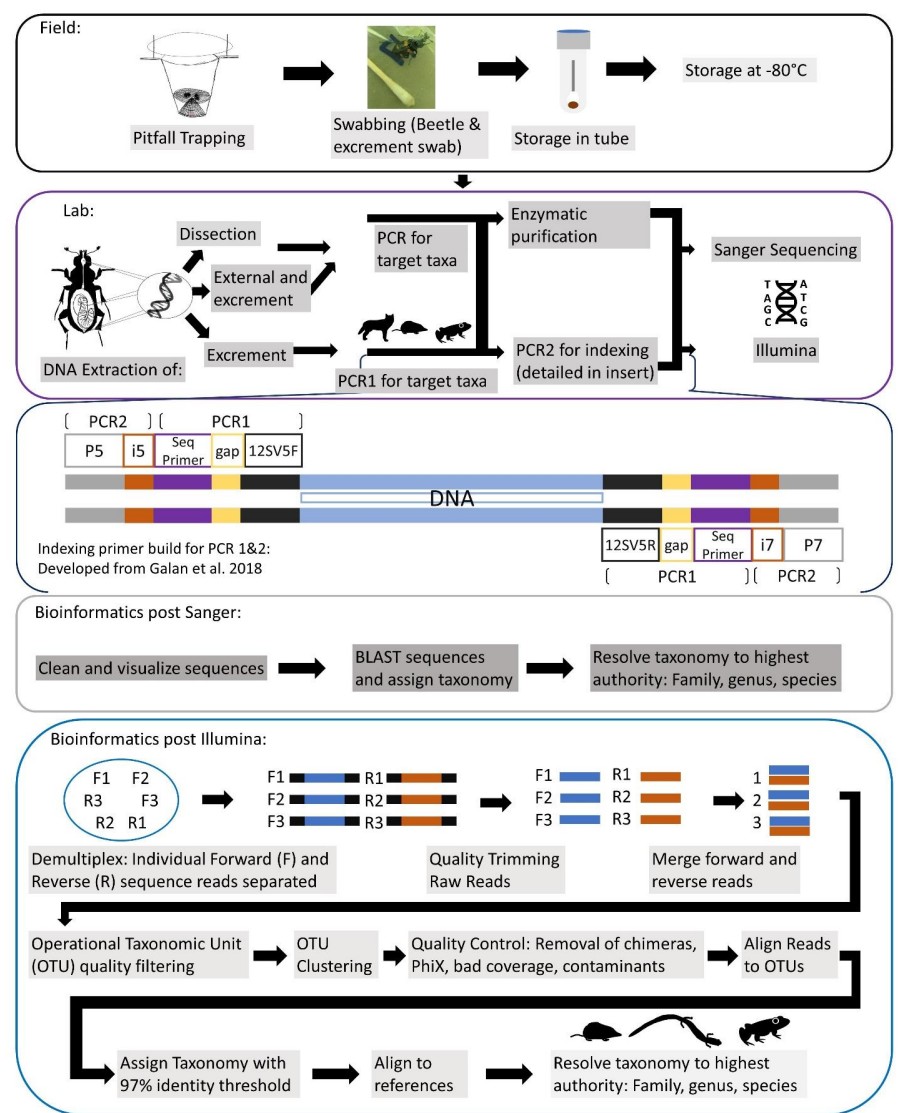

**Figure 2** **Comprehensive workflow of methods to conduct the iDNA study in its entirety from the field to the lab and ending with the bioinformatic pipeline to be followed.** Conceptualization by Katherine Higdon and Melanie Ottino. All photos and drawings contained within created by Katherine Higdon.

TAGAACAGGCTCCTCTAG; 12S-V5-R: TTAGATACCCCACTATGC; (*Riaz et al., 2011*), and 5.65 µL of molecular grade water. We followed thermocycling conditions from *Riaz et al. (2011)* with thermocycling conditions being: 10-minute activation at 95 °C, then 45 cycles of 95 °C at 30 s, 52 °C for 30 s, and 72 °C for 30 s, followed by a final extension step at 72 °C for seven minutes. PCR reactions had an additional negative control with molecular grade water and DNA extraction negative controls. We confirmed successful DNA amplification from PCR with a 1.5% agarose gel. The beetle samples that visually showed amplification on an agarose gel along with two DNA extraction controls were enzymatically purified using ExoSAP-IT™ (Thermo Fisher Scientific) by combining 14 µL

molecular grade water, 0.5 µL of ExoSAP-IT™, 2.5 µL of the 12S-V5F primer, and 2 µL of PCR product for each sample. Thermocycler conditions were 37 °C for 15 min followed by 80 °C for 15 min. The purified product was then sent to Azenta Life Sciences for Sanger sequencing. Sequences were cleaned and processed with Geneious Prime (version 2020.1.1; Biomatters Ltd., Auckland, New Zealand); samples were assessed for multiple peaks on chromatograms (likely representative of multiple diet items) and for poor sequence quality to be removed. We determined each sequence's taxonomic identity using the nucleotide Basic Alignment Search Tool (BLASTn) with the National Center for Biotechnology Information database (NCBI; http://www.ncbi.nlm.nih.gov/BLAST). All sequences were matched at a >98.5% level and samples with similar percent taxonomic identities were determined based on knowledge of local taxa (*Gurtler, 2020*; *Gusick, 2022*).

*Illumina sequencing:* Given we demonstrated the ability to amplify vertebrate DNA and optimized our field sampling protocol, we further tested diet content with metabarcoding methods and high-throughput sequencing on individuals from 2022. Four pooled DNA samples representing a total of 18 *Necrophila americana* (two to six individuals per sample) were selected for testing along with two extraction negative controls and one PCR negative control as all replicates were carried through the same PCR reactions. Each sample had replicated sequencing across the different volumes of AmpliTaq Gold Polymerase used, described below, resulting in a total of eight replicates per sample to determine the lowest volume of AmpliTaq Gold Polymerase needed to for successful species detection. We amplified DNA using primers targeting the 12S rRNA vertebrate gene region with added sequencing primers at the 5′ends (12SV05F: TCGTCG-GCAGCGTCAGATGTGTATAAGAGACAGTTAGATACCCCACTATGC; 12SV05R: GTCTCGTGGGCTCGGAGATGTGTATAAGAGACAGYAGAACAGGCTCCTCTAG; (*Riaz et al., 2011*). We followed metabarcoding library prep as outlined in *Galan et al. (2018)*, which includes PCR1 that amplifies our vertebrate target gene region; a bead-clean with PCR1 product; PCR2 to attach individual barcodes and sequencing indexes to the amplicons from the PCR1 product; and a final purification and normalization for sequencing prep. PCR1 reactions were in 20 µL total, with volumes of 2 µL DNA extract, 0.8 µL, 0.6 µL, 0.4 µL, or 0.2 µL AmpliTaq Gold Polymerase, 2 µL of GeneAmp™ 10X Gold Buffer, 1.5 µL of 25 mM MgCl, 1.6 µL of 5 mM dNTPs, 0.4 µL of the forward and reverse 12SV05 primer pair, and molecular grade water to volume with thermocycler conditions being the same from 2019 described above. PCR product was then purified using 30 µL of Sera-Mag SpeedBeads Carboxyl Beads (Thermo Fisher Scientific, Waltham, MA, USA). PCR2 was conducted as follows: 5 µL cleaned PCR product, 0.2 µL or 0.1 µL AmpliTaq Gold Polymerase, with 2 µL GeneAmp™ 10X GoldBuffer, 1.6 µL of 25 mM MgCl, 0.8 µL 5 mM dNTPs, 1 µL each of i5 and i7 Illumina indexing primers (*Galan et al., 2018*; Fig. 2), and nanopure water to bring it to 20 µL total. Our thermocycling conditions were as follows: 10-minute activation at 95 °C, then 8 cycles of 94 °C at 30 s, 55 °C for 1 min, and 72 °C for 1 min, followed by a final extension step at 72 °C for ten minutes. Lastly, we purified and normalized PCR2 product according to the manufacturer's instructions from Charm Biotech's Just-a-Plate PCR purification kit. Samples were then pooled and

sequenced using an Illumina Miseq Nano kit V3, 2x300 cycle, at Michigan Technological University's Genomic Disease Surveillance Lab.

We used the open-source python-based APSCALE metabarcoding pipeline (*Buchner, TH & Leese, 2022*; https://pypi.org/project/apscale/) to assign taxonomic identity. Steps of the pipeline were that sequence reads were trimmed and demultiplexed on the Illumina instrument by bcl2fastq Conversion Software (v2.20, https://support.illumina.com/) using the fusion primer trimming adapter and unique barcode combinations to obtain forward and reverse read files for all Silphid iDNA samples. Paired-end read merging, primer removal, quality filtering, and dereplication were implemented with standard parameters, detailed in *Buchner, Macher & Leese (2022)* and excluding any merged reads with length <60 bp or expected error rates >1. Operational Taxonomic Unit (OTU) clustering and denoising were implemented in VSEARCH with a 97% similarity threshold and default alpha value ($\alpha =2$). We used the python-based version of R-package LULU with default parameters filtered OTUs to improve biodiversity estimate accuracy (*Frøslev et al., 2017*). Finally, we used the NCBI BLAST and the Assign-Taxonomy-with-BLAST Python script (2020, https://github.com/Joseph7e/Assign-Taxonomy-with-BLAST) to resolve taxonomy to consensus considering a maximum of ten database hits with a percent identity weighted algorithm. Hits only with the default percent sway value (0.5) of the top database hit were considered for determining taxonomic consensus. Default thresholds of the Assign-Taxonomy-with-BLAST were used for taxonomic level identification with a threshold of 97% for species, 90% for family, and 80% for phylum. Any OTU found in only one replicate was not reported due to the likelihood of being a sequencing error.

# RESULTS

## Field results

In 2019, we collected 39 Silphids: 20 Silphid adults and 19 carrion beetle larvae. Silphid beetle species trapped included 15 American carrion beetle (*Necrophila americana*) adults, four margined carrion beetle (*Oiceoptoma noveboracense*) adults, and one tomentose burying beetle (*Nicrophorus tomentosus*). All larvae found in pitfall traps were likely *N. americana* as these individuals are solid black and they are morphologically distinct from other beetle larvae. No beetles were found in traps containing catfish bait and only one tomentose burying beetle was caught in a pitfall trap baited with a pinky mouse. The raw ground beef was most effective as a bait attractant, however we found this messy to work with (*e.g.*, it could easily contaminate the sides of traps as it was being deployed). From the 39 total Silphids, 59 total samples were assessed through the agarose gel step (39 exterior/excrement swabs from adults and larvae and 20 adult dissection swabs), and only samples with visual amplification were sequenced. In 2022, we collected 923 Silphids. Of which, 58 were dead in the trap, and 65 were found in the trap after another animal had removed the mesh. There were 127 margined carrion beetles, 496 American carrion beetles, one banded sexton beetle (*Nicrophorus investigator*), 61 roundneck sexton beetles (*Nicrophorus orbicollis*), three pustulated burying beetles (*Nicrophorus pustulatus*), five tomentose burying beetles, 16 common sexton beetles (*Nicrophorus vespilloides*), and 214 larvae.

## Sequencing results

*Sanger results:* Of the 59 total extracted samples, 16 beetle samples were Sanger sequenced, and two came back with non-specific amplification with two peaks, with blast identities of *Homo sapiens*, and another was low sequence quality and did not match any species (Table S1). The rest all yielded identifiable sequences, and based on our BLAST results, six samples aligned at >98.5% identity with the following: star-nosed mole (*Condylura cristata*), deer mouse (*Peromyscus maniculatus gracilis*), a canid (*Canis* sp.), a Phasianid (which includes chickens, grouse, turkey, and pheasants), and cow (*Bos taurus*) (Table 1). The remaining seven sequenced samples were human. Five successful non-human sequences were from externally swabbed samples where a beetle defecated, and one was from a stomach sample. The two individuals that sequenced *Bos taurus* were found under the mesh in the field (likely bait contamination). All reported samples had high assigned percent identity (>95.79%) reported in BLASTn. The two DNA extraction negative controls that visually showed amplification on the agarose gels were also sequenced and came back with only human DNA. PCR negative controls did not visually show amplification and were not Sanger sequenced. Larvae swabs only came back with human DNA. The success rate of external swab samples from larvae and adults that came back with non-human diet items (12.8% from 5/39 total) is therefore lower compared to the total number of samples for only adult Silphids (25% from 5/20 total), and even lower for dissection samples at 5.0% (1/20). Thus, in 2022 we only collected excrement swabs.

*Illumina results:* All the samples we sequenced on an Illumina platform were American carrion beetles with the associated DNA extraction negative controls and a PCR negative control. Two OTUs identified as human and one identified as a canid in the two DNA extraction negative controls and were classified as laboratory contamination. Replicates of each sample amplified a range of zero to five target vertebrate species with species detection being similar in the same samples between Taq volumes but differing by read count (Table 2). Samples using 0.2 µL AmpliTaq Gold for PCR2 performed the best based on sequence read count and quality (Table 2) and average number of target reads per sample for 0.2 µL of Taq was higher as well at 5,648 compared to 1,582 reads, with more 0.1 µL Taq volumes of PCR2 failing (Table 2). Average number of reads per sample by Taq volume for PCR1 differed at 0.2 (768), 0.4 (3,439), 0.6 (1,898), and 0.8 (1,128). A total of eight unique OTUs were identified in the genera *Anaxyrus*, *Blarina*, *Procyon*, *Peromyscus*, and *Sus* and the families of *Cyprinidae*, and Salamandroidea, with two OTUs both resolving to *Blarina* (Fig. 3). Species detected in only one PCR replicate were removed from reported results as they are most likely sequencing errors.

## DISCUSSION

We successfully extracted, amplified, and sequenced vertebrate DNA from Silphid beetles, demonstrating the utility of these beetles for further iDNA studies for the first time. Between both years, we detected a variety of species including several small mammals in the genera *Blarina*, *Peromyscus*, and *Condylura*, as well as a canid (*Canis* sp.), northern raccoon (*Procyon lotor*), and an individual in the subfamily *Phasianidae* (Fig. 3 & Table 1),

**Table 1  Beetle species ID and associated data for Sanger sequenced diet samples obtained in summer of 2019 including sample type, referring to whether the sequence came from the external field swab with defecation (Field) or from gut dissection in the lab (Dissection).** Bait each individual was attracted to is also noted. Sequence ID determined with the nucleotide Basic Alignment Search Tool (BLASTn) from the National Center for Biotechnology Information database (NCBI; http://www.ncbi.nlm.nih.gov/ BLAST). An asterisk (*) indicates that sequence identity is likely bait contamination.

| Beetle ID | Sample type | Bait | Sequence ID | Percent ID |
|---|---|---|---|---|
| *Necrophila americana* | Field | Beef | *Condylura cristata* | 98.55% |
| *Necrophila americana* | Field | Beef | *Canis sp* | 100% |
| *Necrophila americana* | Field | Beef | *Bos taurus** | 98.57% |
| *Necrophila americana* | Field | Beef | *Bos taurus** | 100% |
| *Necrophila americana* | Dissection | Pinky Mouse | *Phassianinae* | 100% |
| *Nicrophorus tomentosus* | Field | Pinky Mouse | *Peromyscus maniculatus gracilis* | 100% |

showing the potential for Silphids to contribute to biodiversity studies, particularly with regards to small mammals. The star-nosed mole (*Condylura cristata*) is a species that is not commonly detected in mammal surveys because it is primarily fossorial and not usually targeted in box-style, small mammal traps (*Hartman & Krenz, 1993*). Percent identity match of this individual was 98.55%, and given that the star-nosed mole is the only known species of mole whose range is within the study area (*Gusick, 2022*), we are confident with identification of this individual to species. We also detected mice from the genus *Peromyscus* from the individual burying beetle trapped in 2019 as well as from the carrion beetles from 2022. The pinky mice bait was unlikely to be a source of contamination for this detection in 2019 as they are most likely the common house mouse species (*Mus musculus*) sold at pet stores. Additionally, the mesh in the pitfall trap was intact with this individual, unlike the detection of domestic cow (*Bos taurus*) from 2019 (Table 1). While it can be reasonably inferred that the detection of domestic cow DNA resulted from bait consumption, we still report this finding as it demonstrates that the methods used can yield diet DNA. It is plausible that consumption occurred between one and six days before obtaining the diet swab, demonstrated by the absence of bait upon arrival and a fairly ''clean'' trap bottom, making it likely that the individuals in question did consume the beef and the state of the cup suggesting that consumption was less recent. The subsequent sequencing result therefore indicates a true diet item, while artificially introduced to the diet, proving the utility of the method. In contrast, detecting *Peromyscus* in multiple samples that were baited with chicken illustrates that this species was detected with the survey method used. Additionally, this species has been detected through traditional small mammal trapping at the same study sites by a different research group (*Gusick, 2022*). To specifically target small mammals in a trapping area, future work could use more expansive trap configurations, specific baits, or light trapping to target burying beetles over carrion beetles if desired (*Backlund & Marrone, 1997*), given their different mammalian host preferences.

We detected the subfamily *Phassianinae*, which includes domestic chicken as well as wild turkey, grouse, and pheasants, in the 2019 season. Since the primer pair we used has low power distinguishing between avian species (*Xiong et al., 2017*) and given the low

**Table 2  Illumina sequencing data from 2022 from chicken baited traps illustrating percent target vertebrate reads per sample that were not from contamination.** Total reads per sample is inclusive of contamination. Reads are sorted by the volume of AmpliTaq Gold polymerase used for each PCR reaction and represent each of the replicates per sample.

| Sample | PCR1 Taq volume | PCR2 Taq volume | Sample content | Vertebrate detections | % Target vertebrate reads | Total reads per sample |
|---|---|---|---|---|---|---|
| S1 | .8 Taq | .1 Taq | Pool (2 individuals) | 0 | 0.00 | 0 |
| S1 | .6 Taq | .1 Taq | Pool (2 individuals) | 0 | 0.00 | 1 |
| S1 | .4 Taq | .1 Taq | Pool (2 individuals) | 3 | 35.59 | 26171 |
| S1 | .2 Taq | .1 Taq | Pool (2 individuals) | 4 | 38.62 | 13953 |
| S2 | .8 Taq | .1 Taq | Pool (4 individuals) | 0 | 0.00 | 0 |
| S2 | .6 Taq | .1 Taq | Pool (4 individuals) | 0 | 0.00 | 0 |
| S2 | .4 Taq | .1 Taq | Pool (4 individuals) | 4 | 72.65 | 18535 |
| S2 | .2 Taq | .1 Taq | Pool (4 individuals) | 3 | 75.47 | 25352 |
| S3 | .8 Taq | .1 Taq | Pool (6 individuals) | 0 | 0.00 | 0 |
| S3 | .6 Taq | .1 Taq | Pool (6 individuals) | 0 | 0.00 | 0 |
| S3 | .4 Taq | .1 Taq | Pool (6 individuals) | 0 | 3.45 | 29 |
| S3 | .2 Taq | .1 Taq | Pool (6 individuals) | 2 | 97.62 | 42 |
| S4 | .8 Taq | .1 Taq | Pool (6 individuals) | 0 | 0.00 | 0 |
| S4 | .6 Taq | .1 Taq | Pool (6 individuals) | 0 | 0.00 | 2 |
| S4 | .4 Taq | .1 Taq | Pool (6 individuals) | 1 | 43.76 | 7558 |
| S4 | .2 Taq | .1 Taq | Pool (6 individuals) | 0 | 0.00 | 4 |
| S1 | .8 Taq | .2 Taq | Pool (2 individuals) | 3 | 72.36 | 22360 |
| S1 | .6 Taq | .2 Taq | Pool (2 individuals) | 4 | 58.13 | 29927 |
| S1 | .4 Taq | .2 Taq | Pool (2 individuals) | 5 | 38.73 | 48220 |
| S1 | .2 Taq | .2 Taq | Pool (2 individuals) | 0 | 0.00 | 1 |
| S2 | .8 Taq | .2 Taq | Pool (4 individuals) | 4 | 91.71 | 21703 |
| S2 | .6 Taq | .2 Taq | Pool (4 individuals) | 3 | 90.23 | 48017 |
| S2 | .4 Taq | .2 Taq | Pool (4 individuals) | 4 | 71.74 | 75120 |
| S2 | .2 Taq | .2 Taq | Pool (4 individuals) | 0 | 0.00 | 0 |
| S3 | .8 Taq | .2 Taq | Pool (6 individuals) | 1 | 0.03 | 15711 |
| S3 | .6 Taq | .2 Taq | Pool (6 individuals) | 1 | 0.01 | 20242 |
| S3 | .4 Taq | .2 Taq | Pool (6 individuals) | 1 | 3.43 | 146 |
| S3 | .2 Taq | .2 Taq | Pool (6 individuals) | 0 | 0.00 | 0 |
| S4 | .8 Taq | .2 Taq | Pool (6 individuals) | 0 | 0.00 | 21827 |
| S4 | .6 Taq | .2 Taq | Pool (6 individuals) | 0 | 0.00 | 35869 |
| S4 | .4 Taq | .2 Taq | Pool (6 individuals) | 1 | 43.92 | 25899 |
| S4 | .2 Taq | .2 Taq | Pool (6 individuals) | 0 | 0.00 | 0 |

quality of iDNA samples, we cannot conclusively determine species identity with available data. The vertebrate primer pair that we used targets a gene region that similarly does not distinguish among gray wolf (*Canis lupus*), coyote, (*Canis latrans*), or domestic dog (*C. lupus familiaris*), all of which are present in the immediate vicinity of our study site (within the County). However, gray wolves were detected on trail cameras deployed in a separate study which ran simultaneously with our iDNA sampling (*Gurtler, 2020*), suggesting we

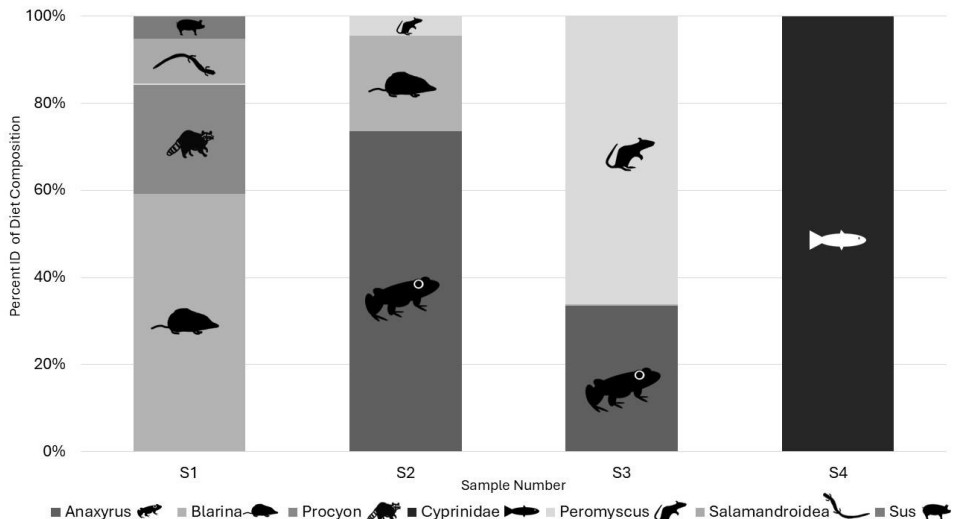

**Figure 3 Percent of target diet OTU identity per individual sample across replicates from Illumina sequencing data in 2022.** Identified to genus in all cases except for *Cyprinidae* and *Salamandroidea* which are identified to family. All figures and drawings created within by Katherine Higdon and Hunter Higdon.

may have detected the presence of wolves *via* our iDNA sampling. Additionally, carrion beetles are known to visit various types of fecal matter in addition to carrion. With proximity of trapping locations to the North Country Trail and a roadside park with frequent visitors that bring pets, it is possible that the canid and human detections could also be from fecal consumption by Silphids. Given the limitations of primers generally used for low quality diet DNA, it is recommended if budgeting allows, to use multiple primers targeting different regions of the vertebrate genome to increase taxonomic identification (*Xiong et al., 2017*). Thus, future work attempting a more comprehensive biodiversity survey should consider an additional primer pair targeting a different gene region (*Boessenkool et al., 2012*).

With consideration regarding the inclusion of larvae in analysis, observational data from 2022 saw small larva individuals being able to fit in mesh holes while intact, therefore increasing the likelihood of contamination through bait consumption. Additionally, carrion beetle larvae are highly mobile and are able to persist for at least seven days without consuming any carrion and will consume other insect larvae including that of their own species if a food source is unavailable (*Trumbo & Dicapua III, 2021*). As such, larvae can be included in beetle community population data, but we recommend they not be included for iDNA analysis due to the high risk of contamination from bait unless it can be guaranteed that the barrier to the bait being used in a study truly keeps them from the bait. All larvae included in Sanger diet analysis only yielded human DNA which was likely lab or field contamination. It is important to note that even the most sterile sampling methods cannot be completely sterile in the field. Multiple steps of field work followed by multiple lab steps all contribute to a higher risk of contamination being detected from human DNA in iDNA studies due to the multiple opportunities for human DNA to be introduced as a
higher quality sample within the obtained diet sample. Consequently, primer binding on contamination becomes more likely, which can even be sequenced in negative controls due to the more sensitive nature of iDNA studies and the employment of shorter primers which are more likely to bind to higher quality and more abundant DNA (*Calvignac-Spencer et al., 2013*; *Galan et al., 2018*; *Drinkwater et al., 2021*).

Sequence success of target vertebrate species from Sanger sequencing was low when considering all individuals. However, if you specifically look at successful amplifications of target diet DNA for adult Silphid field swabs at 25%, it falls within the typical range of iDNA sequencing success (*Calvignac-Spencer et al., 2013*) and are similar to the rate of detection in some more modern studies (*Lynggaard et al., 2019*; *Drinkwater et al., 2021*). Comparing the two sampling types for Sanger sequencing, our results suggest that exterior swabs of exoskeleton and excretion samples were more successful compared to gut dissection, which could potentially be due to excretions expelling any remaining viable diet DNA prior to dissection. However, since it was common for Silphids to produce excrement quickly after being handled, using only gut dissections likely would not produce as many results as collecting the excrement would. It is worth noting that studies with iDNA can have variable success rates, ranging from 21% to 100%, with the 100% success rate observed in samples where blood feeding had been witnessed (*Calvignac-Spencer et al., 2013*). Our low success rate of sequencing non-human diet DNA with Sanger could be due to a combination of (1) potentially accelerated digestion in the beetles' stomachs, relative to other insect species, due to a mutualistic gut microbiota that accelerates digestion to allow the beetles to consume carrion before microbial growth can occur on the carrion as well as to assist in competition for carcass control (*Vogel et al., 2017*), and (2) our method of checking traps a week apart. It is possible the beetles were in the pitfall traps for multiple days prior to acquiring the sample, and that prior to being trapped they had not recently had a meal, potentially reducing viability as their stomach contents could be digested quickly (*Vogel et al., 2017*). Based on this result, we checked traps more frequently in our second field season. Our success rates for non-human diet items are comparable to a similar study of novel iDNA which sampled large-bodied, individual dung beetles in the genus *Catharsius* in groups of six and small-bodied dung beetles pooled as a community to determine the success of and application of using dung beetles for iDNA in Southeast Asia (*Drinkwater et al., 2021*). Their use of pooled samples and high throughput techniques found 12 detections within 12 different samples of pooled beetles representing six mammalian species but did not clearly state the exact total number of beetles sampled (*Drinkwater et al., 2021*). Our Illumina sequencing results had a higher success rate, with all reported species detected in at least two of the replicates, and all samples finding at least one unique target vertebrate OTU. The higher success rate of the second season compared to the first is in line with typical iDNA success and could be due to checking traps more frequently in addition to also using high throughput sequencing methods.

This higher success rate of external swabs compared to dissections influenced the decision to employ only excrement swabs in the second season. The other factors determining the use of only excrements for the 2022 season were due to the messier nature of using PBS buffer in the field to help with the ease of external swabbing as well as the possible negative

effects the buffer may have on the insects. The PBS buffer is known to cause irritation to eyes, skin, and respiratory function in humans (*Fisher Scientific, 2000*). For 2022, since we wished to conduct the study on a larger scale with minimal negative ecological impact, we employed live trapping and releasing insects after swabbing excrements, painting with a bioluminescent, non-toxic paint powder. To ensure we were assessing diets of Silphids, we used the dry swabs in the field and only obtained excrement samples as external swabs have the potential to detect species that beetles came in contact with, not just those they consumed. This type of catch and release study is not unprecedented, as a study by *Kerley et al. (2018)* trapped *Circellium bacchus*, a flightless dung beetle, for a diet iDNA study where individuals were released after a fecal sample was obtained from each beetle.

Based on our Illumina results, we concluded that 0.4 μL of AmpliTaq Gold in PCR1 could successfully capture the potential biodiversity while still being cost effective. Additionally, our results reported in Table 2 indicate that the use of 0.2 μL of AmpliTaq Gold polymerase in the second PCR reaction were more successful, as there were sequencing failures in multiple samples with lower Taq volumes in PCR2, as indicated by samples with 0 vertebrate detections. The specific duration of time for mammal DNA to be able to be retrieved after feeding events should be further explored for multiple Silphid species, including both carrion and burying beetle groups; given that we are aware of the abilities of microbial and fungal communities in the guts of some of these species to accelerate digestion (*Vogel et al., 2017*).

Different bait recommendations for trapping Silphids range in the literature from ground beef or cubed chicken, catfish bait, pinky mice, cheese, etc, and targeted studies would most likely determine the preferred type of bait used to both limit DNA contamination, but also be most cost effective and easiest to replace in the trap without contacting sterile surfaces. For instance, we found the catfish bait, recommended in *Andrew (2016)*, was harder to install in traps without touching the sides of the plastic cup. Furthermore, not knowing the specific recipe of ingredients in the bait could limit determinations of bait contamination when assessing diet. *Andrew (2016)* found that baits of rotten meat all performed better than the artificial baits, which include catfish bait (2016), which was reflected in the side-by-side comparison where no individuals were found in the catfish bait traps with all reported captures occurring in the beef, mice, and chicken traps. Another reason to switch to chicken as bait rather than pinky mice for the 2022 field season was to avoid the possibility that any small mammal detection would be construed as bait contamination. The final reason for ultimately switching to chicken over pinky mice or ground beef was due to the cost and cleanliness as well as the ease of accessibility to obtain the bait.

There were also instances in which the mesh was found outside of the trap after a week, most likely from animal disturbance such as raccoons attracted to the bait. At these traps, when the plate covering was ripped, or the mesh was found on the ground next to the trap, animal disturbance was assumed. Individual beetles were still in some traps in the ground but were not sampled for diets for the second field season due to high potential for contamination. Beetles are less likely to be lost to animal disturbance if traps are also checked more frequently, as was done in the second sampling season. It is a possibility that human DNA from fecal matter or other sources is truly a part of Silphid diets, but

contamination is more likely the cause of detection. Contamination can be determined through sequencing negative controls and removing overlapping sequences from results. Additionally, sequencing replicates can help identify false sequences generated from non-specific primer binding (*Calvignac-Spencer et al., 2013*).

It should be noted that there were more traps deployed in the first field season, 216 as compared to 151 in the second season. This difference was due to the fact that the 2019 field season deployed traps in pairs rather than single traps as the 2022 season did, allowing for traps to be doubled in the 2019 season as pairs of 108 traps rather than 151 individual traps. Since each trap pair had two different baits in 2019, use of only chicken in 2022 eliminated the need for a second trap. Despite this difference, there were markedly more individuals caught in the follow-up year. One possible reason for the catch success could come from growth in years post silvicultural harvest. While the exact trapping treatments are not related to the breadth of this study, as the intention is to establish Silphids as an iDNA tool, the fact that the preliminary data came from a site two years post-harvest compared to the same site five years post-harvest most likely influenced the vast difference in trapping success. Other invertebrate indicator species such as beetles in the family Carabidae (ground beetles), take multiple years post-harvest to recover in population size and density (*Moore et al., 2004*).

Since diet DNA could be obtained with Silphid excrements, using a mark-recapture framework with these beetles could use iDNA methods without requiring euthanizing individual beetles. This technique would have potential application to the American Burying Beetle (*Nicrophorus americanus*), recently downlisted to threatened status (*U.S. Fish and Wildlife Service, 2019*), as an additional method to determine potential host preference to aid in restoration and reintroduction efforts.

## CONCLUSIONS

Use of Silphid iDNA may serve as another "tool in the toolbox" for wildlife biodiversity or monitoring surveys, targeting small mammals that are difficult to trap, or in situations when noninvasive sampling is preferred, such as with rare or vulnerable species. While high throughput sequencing methods are more typical of iDNA studies, Sanger sequencing can still be a valuable tool to establish the viability of a family group of insects for future studies, as has been done before. Sanger sequencing was used for the first portion of this work due to the small sample size and faster digestion in the beetles that would suggest a low possibility of there being two diet species present. The combination of Sanger sequence generated data in conjunction with test run results using high throughput methods on Silphid diets validate the method, while further studies are needed to evaluate the potential of the method when applied to a larger scale study. Additionally, a lower volume of AmpliTaq Gold can be used for sequencing on an Illumina platform than what was originally used with Sanger sequencing methods, thus reducing lab costs. Further sampling can explore if Silphids iDNA studies can resolve mammalian communities and provide potentially deeper insights into faunal presence or mortality studies. We recommend checking traps as frequently as possible and using multiple primers targeting different genome regions

if funding allows. Additionally, to specifically target diets, collecting excrement without swabbing beetle exoskeletons is recommended. Ultimately, here we have confirmed that vertebrate DNA can be successfully sequenced from Silphid samples.

## ACKNOWLEDGEMENTS

This research was conducted on homelands and ceded-territory, defined by the Treaty of 1842, of the Ojibwa (Chippewa) people. We would like to thank A. Schoch, S. Lane-Clark, and K. Bershing for assisting with trapping efforts. Additional thanks to S Hervey, T. Barnes, and T. DeGroot for assisting with lab work. We would also like to thank the Ford Center Forest and NHSEED project (Dr. Y. Dickinson and Dr. C. Webster). Thank you also to C. Külheim for help with manuscript review.

### Funding

The funding support for this project was provided by the Ecosystem Science Center, Pavlis Honors College (Summer Undergraduate Research Fellowship), and College of Forest Resources and Environmental Science at Michigan Technological University. Additional funding support was provided by the United States Department of Agriculture- National Institute of Food and Agriculture (No. 2017-67013-26261). The funders had no role in study design, data collection and analysis, decision to publish, or preparation of the manuscript.

### Grant Disclosures

The following grant information was disclosed by the authors:
The Ecosystem Science Center, Pavlis Honors College (Summer Undergraduate Research Fellowship).
College of Forest Resources and Environmental Science at Michigan Technological University.
The United States Department of Agriculture- National Institute of Food and Agriculture: 2017-67013-26261.

### Competing Interests

The authors declare there are no competing interests.

### Author Contributions

- Katherine E. Higdon conceived and designed the experiments, performed the experiments, analyzed the data, prepared figures and/or tables, authored or reviewed drafts of the article, and approved the final draft.
- Kristin E. Brzeski conceived and designed the experiments, analyzed the data, authored or reviewed drafts of the article, and approved the final draft.
- Melanie A. Ottino analyzed the data, authored or reviewed drafts of the article, and approved the final draft.

● Tara L. Bal conceived and designed the experiments, analyzed the data, authored or reviewed drafts of the article, and approved the final draft.

### DNA Deposition

The following information was supplied regarding the deposition of DNA sequences:

The sequences generated in this project matched previously published GenBank samples: OL521838.1, KX754488.1, MZ433367.1, MW423198.1, LC082227.1, JN393210.1.

### Data Availability

The raw qPCR data, filtered read numbers, and OTU sequences and hit table are available in the Supplementary Files.

### Supplemental Information

Supplemental information for this article can be found online at http://dx.doi.org/10.7717/peerj.17636#supplemental-information.

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
