# Peer review of "Establishing Silphids in the invertebrate DNA toolbox: a proof of concept"

_PeerJ, doi:10.7717/peerj.17636_

## Round 0.1 · original submission · Major Revisions

Dear Dr. Higdon and colleagues:

Thanks for submitting your manuscript to PeerJ. I have now received two independent reviews of your work, and as you will see, the reviewers raised some relatively minor concerns about the research. This is great and indicates optimism for your work and the potential impact it will have on research studying silphid biology and ecology.

There appears to be more that can be done regarding clarity in the Introduction as well as the Materials and Methods, although the writer should be improved throughout the manuscript. The figures and tables and their legends also need to be clearer.

There are many minor suggestions made by both reviewers that need to be addressed in your rebuttal.

Therefore, I am recommending that you revise your manuscript, accordingly, considering all the issues raised by the reviewers.

Good luck with your revision,

-joe

Reviewer 1 ·

Basic reporting

Language is clear and professional, but literature cited is at times outdated for this growing field (see below). The study itself is valuable and fills a gap as the authors state. Please see my comments regarding the figures:
Fig 1. Please refine this illustration. Multiple strokes should be consolidated into vectors. Please also add labels corresponding to the different elements of the trap you describe in lines 105-110 (cone, bait, mesh, etc.). As it stands it is difficult for the reader to accurately replicate the trap design.
Fig 2. Please modify the color scheme for this figure. The shades of grey are too close to distinguish from one another or to draw any conclusions from the data. Please consider incorporating patterns to differentiate across genera. Please also label the axes.

For the tables, please also report the percent identity from your blast search with illumina sequences.

Experimental design

It's clear that much effort went into the field work and sampling design of this study. However, as it stands, it's difficult to draw conclusions about the most effective bait from this data set. At times it is unclear why certain decisions have been made (see below). Clarifying the reasoning for each methodological decision will improve the quality of the manuscript.

Validity of the findings

I worry that swabbing the beetle exterior is not necessarily indicative of its latest meal but rather the various species it encounters in its habitat. This is still useful information for species monitoring but I would hesitate to draw conclusions regarding beetle feeding behavior or preference based on these results. I recommend making this distinction early in the manuscript and treating the swabs and dissections as two different data types with two separate conclusions.

How did you resolve parsing bait contamination from your diet DNA?

If your conclusion is that swabbing carrion beetles is more effective than analyzing gut contents, how do you manage bait contamination from their exteriors?

Additional comments

Abstract, line 17: analyses is should read analyses are
Abstract, line 29: can you assume diet DNA from an external swab? I think a more appropriate claim would be environmental DNA detection broadly sourced from external swabs and specifically diet DNA sourced from gut dissections (iDNA). Making this distinction early on in the introduction (line 37) will help maintain clear separation of the methods and conclusions.

Line 57-58: This statement is inaccurate because iDNA is not limited to blood feeding insects. Blood feeding leeches appear to show a similar trend towards detections of small mammals. Please support this statement with more contemporary citations, there have been numerous iDNA studies since 1998.

Methods
Lines 113-114: Are there known effects of seasonality in terms of host species present in the sampling locality?

Line 107: How does this contribute to contamination of the gut contents at the time of dissection?

Line 128: Are there data to suggest burying beetles prefer pinky mice? How was this decision made?

Line 130: Can you draw any conclusions on the efficacies of the various baits? Was chicken the most effective bait? Why was chicken selected for the entirety of 2022 sampling?

Lines 147-149: Consider including your dissection protocol here, on first mention of dissections in your methods.

Line 149: What did the results from 2019 show and why was the decision made to only collect regurgitate while in the field? Why was exoskeleton swabbing excluded? Please be explicit about your decision making.

Line 162: I assume due to their small size? Do larvae also feed on carcasses? It should be made clear why exactly larvae were excluded from your analysis. This is the first mention of larvae in the manuscript. It would be interesting to learn more about how their behavior differs from beetle behavior.

Line 173-174: How many samples were processed for each year, total? How many negative controls total?

Line 178: replace “as such” with “therefore”

Line 231-232: What was your “percent sway” when assigning taxonomic identity? How did you justify this value?


Results
Line 260-261: What are your percent identity cutoffs for your Sanger sequences? What do you consider a “high” assigned percent identity? Please justify your reasoning.

Line 282: How can you be confident the Phasianid is not bait contamination?

Line 286: “deer mice…are of preferable host size for burying beetles…as opposed to carrion beetles…”

Line 287: What are their reproductive strategies? This is not apparent to the reader, please elaborate.

Line 302: What distinction do you make between iDNA and diet DNA?

Line 307: This is the first mention of camera trap data. Please clarify earlier in the ms (in the methods) if every pitfall trap was deployed simultaneously with a trail camera.

Line 323: This is a bit unclear, do you mean the community population data of the beetles themselves?

Line 332: How do you define success? Do you mean a successful amplification? What exactly are you observing at 25%? Please rephrase and elaborate.

Line 387-388: Can you elaborate why a different number of traps were deployed in different seasons?

Line 397: This is a good idea but feels like a weak suggestion after concluding that swabbing was more effective at detecting target taxa than diet DNA

Reviewer 2 ·

Basic reporting

I think overall this is a good proof of concept study and the authors have clear objectives. In general, I think there could be an improvement in the clarity of the writing throughout the whole article. While I appreciate that the authors were trying different methods, it is hard to follow whether samples were subjected to which sequencing treatment and how the PCRs were conducted etc.

The language could be more consistent throughout when refering to sampling method for example - sometimes swabbing is referred to, or faecal, or regurgitate, but I think these are one technique and they were not distinguished between. Perhaps including a table or adding a diagram with sample numbers showing how many individuals were subjected to which treatment would be very helpful for the reader. The results section is also lacking some basic information, especially when reporting the NGS results as I think there could be more information regarding the basic on the sequencing results, e.g. numbers of reads, reads in the negatives etc.

The figures and tables need more work, are figure two and table 2 from which year? There needs to be greater description in the legends. It is also not clear if the sequence data has been deposited online to NCBI.

Experimental design

For a short proof of concept I think the experimental design is sound, because of the nature of the study, I think it is understandable that some techniques were tried and modified. Especially in the field. However, I could not establish if multiple PCR replicates were conducted - I think this is quite important in a metabarcoding study with only one marker and perhaps just needs to be highlighted better in the reporting.

Validity of the findings

There is no indication of how well the OTUs match to the database or what cut-offs and thresholds were used. The supplementary just includes read counts. Is this only 100% matches with the database? How was it decided to only restict the match to genus or family? Also in the supplementary there is only shortened latin names. I think the full identification needs to be reported.

Additional comments

Introduction
44-45. Might be best to put the references in order here - or with the species they refer to
47-49. This is not so clear - initially sounds like commonly all the beetles called burying beetles, but then from line 48 this feels like you are saying that only burying beetles specialise on small (mammal?) carcasses.
47. I think you should specify small mammal carcass. Small carcass could just be a carcass of a large mammal that is small.
64. Less particular than what? The Nicrophorus genus? I think this section needs a little work. Its quite confusing.
73. Mortality information of what? The carcass? I think this needs expanding a little, sounds like it could be the mortality of the beetle.

Methods
122-123. Do you mean here something that was very different from the local fauna so you are able to identify it? Or physical reduction in the amount of contamination?
123. Catfish has previously been shown to work?
127. Because you see an increase in Nicrophorus with the pinky mice? Is there a reference for this?
133. Just individuals in Michigan have clear differences? Not across the whole range?
149-151. It is not clear what the results from 2019 are (yet). I think you need to reorder this section. In 2022 you did not conduct dissections or swabbing? You only collected regurgitates from the individual plastic bags? Regurgitate and faecal samples?
154-155. I think you could be a bit clearer/consistent in the language referring to the sampling method
162. Did you swab the larvae? Or extract the whole larvae?
192. How many samples was this?
196. Was this just for 2022 samples? Could you specify?
202. Taxa specific primer? How do you know which taxa you are targeting as its a mixed sample? Do you mean target region?
202. Were only single PCR replicates per sample conducted?
202. PCR2 is attaching barcodes to the amplicons from PCR1? For multiplexing? So PCR2 is an indexing PCR only?
219-222. I think you would demultiplex, trim and process your sequences before identifying the taxonomy? So I think this first sentence need to be moved down. Or do you mean that within the APSCALE pipeline you conduct all the steps below? If this is the case, I think you need to clarify that it is not just taxonomic identification but read processing up to the identification of the unknown amplicons. Which I am not sure it is as you go on to say you identify the species using BLAST.
226. (2022)? Is this from the reference above?
227. Did all OTUs match at >97%? What threshold did you use to assign genus or family?

Results
236. The larvae are carrion beetles but the adults are not?
244. These are very different numbers between 2019 and 2022.
246. The latin names here are inconsistent, I think you should put them all or put all the common names
253. How many samples is “the rest”
254. Aligned? How well did they match? This is a list here of species, subspecies, a genus and a family
258. In the methods it sounds like you cannot tell whether you are swabbing the faeces or the regurgitate. Maybe stick to the same language here.
259-261. What trap bait was it? If they were found under the mesh, and this was beef, this is very likely to be contamination and should be disregarded. Did you run PCR negatives with the Sanger sequencing? Can you report the results here as well? Was there any amplification?
264. Its not clear here what you are comparing against? Where are the results from the dissections? Is this the “one stomach sample” from line 259? As this is the reason you are using to make the case for just swabbing in 2022 I think you need to be clearer that the success rate for the swab (both faecal and regurgitate I assume) is much greater than the gut dissections. Also you should probably elaborate on this when you first introduce it in the methods.
266. Why did you not sequence any of the other species?
267. I think this needs more information. How many reads did you get? How many OTUs did you identify? How well did the OTUs match the database? How many reads were in the negatives?
267. What is this (2)?
267. Was this across PCR replicates?
268-269. This could be clearer and elaborated on - do you mean between 0-5 vertebrates were detected per PCR replicate? Individual? Pool? Its not clear.
270-272. This section is really not very clear and quite hard to follow. What comparison are you making here? What are these numbers 0.2, 0.6 etc? I think these are in relation to the Taq in PCR1. Is this after cleaning with LULU? These are non-unique OTUs?
274. One PCR replicate? Or in one sample? Or in one beetle? Even if the OTU was well supported? Perhaps this represents a rare species in the area

Discussion
279. This is not the first use of beetles for iDNA, I think you mean Silphids. Additionally I think you should include this paper in the discussion - as they sequenced DNA from beetle faecal samples (Kerley, et al. 2018, Diet shifts by adult flightless dung beetles Circellium bacchus, revealed using DNA metabarcoding, reflect complex life histories. Oecologia, 188, 107-115)
281. A canid? Is this considered a small mammal?
284. Are you identifying these to species or genus? In line 281 this is just referred to as the geus,
286-288. Do you detect the deer mice in only the burying beetles? You need to make the link here
Give the species for the deer mice - so you can compare to the pinky mice species
290-291. Can you clarify this sentence - the mesh of the trap. Which individual are you referring to and where was the Bos detection
293. What is the feeding ecology of the beetles? How often do they need to feed and what would be a plausible feeding turn over of DNA in their guts?
290-292. Can you be consistent with using either common or latin names.
304. Primer region? Do you mean the target region of the primer?
307. This is interesting - can you make other comparisons with the camera traps? Not necessarily a full analysis but to give some sense of whether the species you have detected are reasonable for the diversity of the region.
318-325. This section feels a bit out of place. There has not been any introduction to the collection or identification of larvae. Or that the larvae were sent for Sanger sequencing.
327. I do not follow what “Multiple steps and location of work” means. In the field vs. the lab? Or geographic location? I would also assume that contamination is introduced because NGS is a much more sensitive sequencing technique as well.
334. I think you need to highlight that the issue with the exterior swab is that you can not tell exactly where that DNA is coming from vs. the gut dissection where you know for sure it is from feeding.
335. Did you make any observations on whether the gut was full? Can you see this? Targeting beetles with full guts would maybe increase the success rate.
345. Is this the reason you got so many more samples in the second year – this is not clear in the methods
250. 12 individuals in 6 mammal species? I think there is something wrong with this sentence. Do you mean 6 OTUs from 12 beetles?
252. All reported species?
363-365. I do not follow. Are the gut communities special in some way? Compared to other species of beetle? Or is this just digestion rate or metabolism
371. Do you find evidence of contimaination from catfish?
372-372. This does not make sense, did the baits have a specific recipe of ingredients?
380-385. This paragraph is a mix of sentences with unconnected points.
387-388. Mistakes in the numbers here - do you mean 2016 to 2022?
400-402. I think you need to also address potential for overharvesting? Especially if you have protected beetles in the area. Could this be a problem? A downside to iDNA sampling in this way.

Table 1. If its the first time for the species name it should be the full latin name - what is P. maniculatus gracilis. Were these 100% matches?

Table 2. I dont think you need to go to 3 decimal places. Also what is this a % of, All samples or within pools?

Figure 1. should maybe be improved a little and could be drawn in a more technical way perhaps - in addition some kind of work flow could be added here to demonstrate how different samples were treated?

Figure 2. You cannot distinguish between the greyscale, can this either be in colour or with different patterns?

---

## Round 0.2 · Minor Revisions

Dear Dr. Higdon and colleagues:

Thanks for revising your manuscript. The reviewers are mostly satisfied with your revision (as am I). Great! However, there are a few issues still to entertain. Please address these ASAP so we may move towards acceptance of your work.

Best,

-joe

Reviewer 1 ·

Basic reporting

Thank you for the changes you've made to the manuscript, they are to my satisfaction.

Experimental design

No comment

Validity of the findings

No comment

Additional comments

No comment

Reviewer 2 ·

Basic reporting

I believe that the authors have largely addressed my initial concerns and improved the manuscript. But there are still a few parts where the clarity of the writing could be improved and the reasoning behind the decisions should be elaborated. I think the new figures are useful but could also be improved by making them more clear and consistent. I have added details below. The article also needs to be checked for the use of the correct symbols - such as the micro-litre and degrees C when writing out the PCR method.

Experimental design

No comment

Validity of the findings

I think the bos contamination of the two sanger sequences needs to be highlighted in the table - it is very likely that this is the cause although this can not be proven 100%. Additionally in the discussion it needs to be clearer that you cannot strictly assess the diet from the external swabs alone.

Additional comments

Line 161 – 163 – You describe the two collection methods used – first method is exterior swab and second is gut dissection. Can you make it clear from 2022 which of these two methods you used?
I also still do not like starting this sentence with “Due to the results from 2019” as we don’t know what those results actually are – I think the section would be clearer if you start with the method and then say this is because our results from 2019 show that the second methos i.e. gut dissection technique was unsuccessful….
Might also be good to refer to Figure 2 here as well

Line 199 – 16 beetles were positive out of how many samples total? In fact, it would be good to get the sample numbers here somewhere

Line 202 – How many of the samples with came back with multiple peaks? Is this something you would expect or are the beetles only feeding on one carcass at a time? This point could be added into the results

Line 204 – Were only 100% matches considered? What were the parameters used here?

Line 212 – Were these pools of the same silphid species? You were expecting different mammal hosts for different species of silphid or at least different sized hosts

Line 214 – Its not very clear why you were modifying/testing the different volumes of Taq, what was the hypothesis? Is it simply to optimise the PCR or did you think that you would increase the detection rate

Line 255 – 268 – Can you be consistent in whether you are writing out the number in full or not, normally below ten is written out

Line 279 – I think that this is high likely to be contamination and at the very least you cannot tell. I think you need to highlight this in table one – as it currently looks like a true result

Line 289 – 290 – Is this because you made the decision not to sequence the other beetle species and these were the most abundant? It would be nice if this information was in the methods, that the pools were only this beetle type? As my first thought when reading about the beetle diversity is how does the beetle species relate to the mammal ID which now is not an issue as there is only one species sequenced.

Line 292 – Was there consistency in the vertebrates within replicate? Were the same OTUs identified from the same sample?

Line 295 – Does the increase in reads come with an increase in detection or quality of detection?

Line 307 - Space missing between Figure 3 & Table 1

Line 311 – Is this percentage match low for your hits?

Line 322 – 324 – Why do you specify between one and six days? Is this the rate of digestion for the beetles? You should be more explicit here. I think though it is plausible to report the result, it is highly likely contamination, especially given that the other two samples with cow bait retrieved different mammals. I think you need to clarify this in the table as well.

Line 342 – This statement should be made earlier as this is also likely the route of the detection of the large domestic mammals as having carcasses in the fields would be highly unlikely

Line 361 – 364 – This sentence should be rewritten to be clearer. It is not that the primers preferentially binds to contamination, that doesn’t quite make sense, the primers don’t know what is and isn’t contamination and you can have low quality contamination. It is more the quality and abundance of the human DNA is likely to be high and therefore more likely to be amplified.

Line 367 – 368 – Typical range? From other studies?

Line 377 – 379 – Maybe it’s the earlier statement that needs to be clarified but I was understanding that digestion would take several days (1-6 days), but here rapid would indicate hours maybe and this is the case in some beetle species i.e dung beetles

Line 382 – Yes, I think this is the main reason – a week apart is a long time for digestion to continue

Line 395 – This is the information that is missing from earlier in the paper about methodological decisions

Line 400 – This sentence is too long and confusing. Can you clarify?
Also you are not “truly assessing the diets” with external swabs – in fact you cannot say for sure that the beetle was using the vertebrate a diet item – however, this is true for excrement swabs.

All the volumes throughout need a microlitre symbol and not a “u” and all temperatures need a degrees sign

Table 2 – are all the traps with the same bait?
Table 2 – what are columns PCR 1 and 2? I would change this to just say “Taq Volume PCR 1…” and % should only be to 2 decimals to be more concise

Figure 2
- Vectors should be aligned, spacing and shapes should be consistent – the boxes are all a bit different
- The swabbing picture is very unclear what it is – a beetle? Or a regurgitate?

Figure 3 - Y axis needs a better label

---

## Round 0.3 · accepted · Accept

Dear Dr. Higdon and colleagues:

Thanks for revising your manuscript based on the concerns raised by the reviewers. I now believe that your manuscript is suitable for publication. Congratulations! I look forward to seeing this work in print, and I anticipate it being an important resource for groups studying silphid biology and ecology. Thanks again for choosing PeerJ to publish such important work.

Best,

-joe

Reviewer 2 ·

Basic reporting

Thank you addressing all my comments and making the revisions. I think that the paper reads alot clearer now and that it will be a useful proof of concept paper.

Experimental design

No comment

Validity of the findings

No comment

Additional comments

No comment